# Vanish into Thin Air: Cross-prompt Universal Adversarial Attacks for SAM2

**Ziqi Zhou**[1,2,3*], **Yifan Hu**[1,2,4,5†], **Yufei Song**[†], **Zijing Li**[‡], **Shengshan Hu**[1,2,4,5†],
**Leo Yu Zhang**[§], **Dezhong Yao**[1,2,3*], **Long Zheng**[1,2,3*], **Hai Jin**[1,2,3*]

[1] National Engineering Research Center for Big Data Technology and System
[2] Services Computing Technology and System Lab      [3] Cluster and Grid Computing Lab
[4] Hubei Engineering Research Center on Big Data Security
[5] Hubei Key Laboratory of Distributed System Security
∗ School of Computer Science and Technology, Huazhong University of Science and Technology
† School of Cyber Science and Engineering, Huazhong University of Science and Technology
‡ School of Software Engineering, Huazhong University of Science and Technology
§ School of Information and Communication Technology, Griffith University
{zhouziqi,hyf1009,yufei17,lizijing,hushengshan,dyao,longzh,hjin}@hust.edu.cn
leo.zhang@griffith.edu.au

## Abstract

Recent studies reveal the vulnerability of the image segmentation foundation model SAM to adversarial examples. Its successor, SAM2, has attracted significant attention due to its strong generalization capability in video segmentation. However, its robustness remains unexplored, and it is unclear whether existing attacks on SAM can be directly transferred to SAM2. In this paper, we first analyze the performance gap of existing attacks between SAM and SAM2 and highlight two key challenges arising from their architectural differences: directional guidance from the prompt and semantic entanglement across consecutive frames. To address these issues, we propose UAP-SAM2, the first cross-prompt universal adversarial attack against SAM2 driven by dual semantic deviation. For cross-prompt transferability, we begin by designing a target-scanning strategy that divides each frame into k regions, each randomly assigned a prompt, to reduce prompt dependency during optimization. For effectiveness, we design a dual semantic deviation framework that optimizes a UAP by distorting the semantics within the current frame and disrupting the semantic consistency across consecutive frames. Extensive experiments on six datasets across two segmentation tasks demonstrate the effectiveness of the proposed method for SAM2. The comparative results show that UAP-SAM2 significantly outperforms *state-of-the-art* (SOTA) attacks by a large margin. Our codes are available at: https://github.com/CGCL-codes/UAP-SAM2.

## 1   Introduction

Recent advances in deep learning have led to the emergence of large segmentation foundation models [4, 16, 33, 40, 43] with impressive generalization capabilities, enabling object segmentation from unseen images. Among them, *Segment Anything Model* (SAM) [4] can output a class-free mask by leveraging prompts (*e.g.*, points or boxes) to accurately localize target objects. Despite its strong segmentation ability, SAM is limited to images. Therefore, SAM2 [28] is recently proposed to integrate a memory mechanism to store features of previous frames, extending SAM to general-purpose video segmentation. Given a prompt (typically on the first frame), SAM2 can continuously track and segment the target object across subsequent frames [11, 32, 49].

39th Conference on Neural Information Processing Systems (NeurIPS 2025).

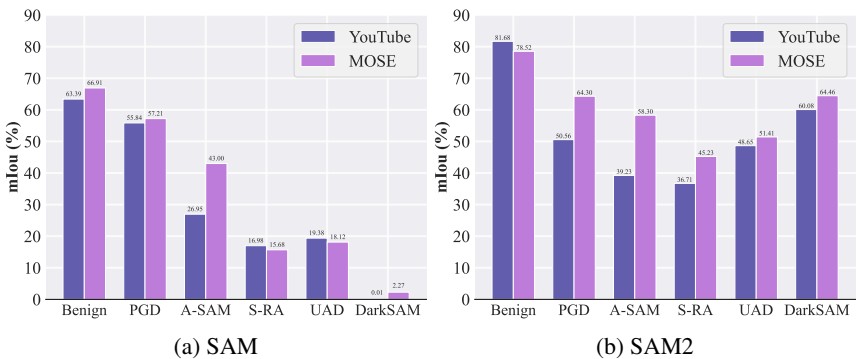

Figure 1: An empirical study on the transferability of existing SAM attacks to SAM2

*Deep neural networks* (DNNs) are known to be vulnerable to adversarial examples [34, 35, 46], where imperceptible perturbations lead to incorrect predictions. Recent studies show that such perturbations can significantly impair the ability of the SAM to segment target objects. Attack-SAM [42] applies PGD [23] to generate sample-wise perturbations for each image. DarkSAM [48] introduces a spatial-frequency universal attack framework, crafting *universal adversarial perturbations* (UAPs) [30, 44, 45] that generalize across diverse images. Given the prompt sensitivity of SAM-based models, recent works [14, 21, 48] also design adversarial examples with cross-prompt transferability. However, despite their effectiveness on SAM, the robustness of SAM2 against these attacks remains unexplored.

Motivated by existing video attacks [36, 37], we investigate whether existing attacks designed for SAM can be directly transferred to SAM2. We evaluate several representative methods, including PGD [23], Attack-SAM [42] (A-SAM), S-RA [29], UAD [22], and DarkSAM [48] on the YouTube [41] and MOSE [6] datasets with $\epsilon = 10/255$. As shown in Fig. 1, existing attacks effectively fool SAM but fail to deceive SAM2. For example, DarkSAM reduces the average segmentation performance of SAM by $98.25\%$ relative to its original performance on two datasets, while causing only a $22.26\%$ drop in SAM2. These results highlight the difficulty of directly transferring attacks from SAM to SAM2.

Given the shared design philosophy between SAM and SAM2, we conduct a comprehensive analysis to uncover the underlying causes of their performance gap in Sec. 3.1. To achieve consistent and accurate segmentation across frames, SAM2 stores the user-provided prompt as a persistent, video-specific representation. It also maintains a memory bank that caches semantic features from some of the previous frames. During inference, SAM2 jointly leverages the prompt and memory bank to guide the segmentation of each frame, repeating this process throughout the sequence. Our findings in Sec. 3.1 highlight two key factors behind the failure of existing attacks: (1) *directional guidance from the prompt*, and (2) *semantic entanglement across consecutive frames*. To effectively attack SAM2, we establish two key objectives. First, the perturbation must generalize across diverse prompts to maintain attack effectiveness. Second, since videos contain numerous frames, we aim to craft a UAP rather than sample-wise perturbations that applies to any frame across different videos.

In this paper, we propose UAP-SAM2, the first cross-prompt universal adversarial attack against SAM2 driven by dual semantic deviation. It generates a UAP that generalizes across videos, frames, and prompts, effectively preventing SAM2 from segmenting target objects, making them *vanish into thin air*. For the first objective, we begin by designing a target-scanning strategy that divides each frame into $m$ regions, each randomly assigned a prompt, to reduce prompt dependency during optimization. Moreover, instead of directly attacking the prompt-dependent masks, we disrupt the semantic features produced by the image encoder to improve the cross-prompt transferability. For the second objective, we design a dual semantic deviation framework that optimizes a UAP by distorting the semantics within the current frame and disrupting the semantic consistency across consecutive frames. Specifically, we design a semantic confusion attack to hinder SAM2's understanding of target objects by injecting noise into the semantic space, a feature shift attack to maximize the semantic distance between adversarial and benign frames, and a memory misalignment attack to amplify inter-frame semantic inconsistency by breaking temporal alignment.

We conduct a comprehensive evaluation of UAP-SAM2 and its sample-wise variant UAP-SAM2* across six datasets spanning both video and image segmentation tasks. The comparative experiments demonstrate that our approach significantly outperforms SOTA attacks against SAM2. Additionally,

defense experiments further validate the robustness of our proposed method. Our main contributions are summarized as follows:

- We propose the first cross-prompt universal adversarial attack against SAM2, revealing the vulnerability of video segmentation foundation models. By designing a UAP, our method consistently misleads SAM2's segmentation across videos, frames, and prompts.
- We design a brand-new dual semantic deviation framework that optimizes a UAP by distorting the semantics within the current frame and disrupting the semantic consistency across consecutive frames.
- We conduct extensive experiments on six datasets across two segmentation tasks to demonstrate the effectiveness of the proposed method for SAM2. The comparative results show that UAP-SAM2 significantly outperforms SOTA attacks by a large margin.

## 2 Preliminaries

Given an input sequence of frames $\mathcal{X} = \{x_i\}_{i=1}^N$ and prompts $\mathcal{P} = \{p_i\}_{i=1}^L$, the SAM2 $f_\theta(\cdot)$ predicts segmentation masks $\mathcal{Y} = \{y_i\}_{i=1}^N$ for each frame $x_i$. For a frame $x_i$, a pixel at coordinates $(m, n)$, denoted as $x_i^{mn}$, is considered part of the masked region if its corresponding mask value $y_i^{mn}$ exceeds a predefined threshold of zero. SAM2 consists of an image encoder $\mathcal{E}_{\text{img}}$ that encodes each frame $x_i$ into a feature embedding $F_i = \mathcal{E}_{\text{img}}(x_i)$; a prompt encoder $\mathcal{E}_{\text{prompt}}$ processes the input prompt $p_i$ and produces the corresponding embedding $Q_i = \mathcal{E}_{\text{prompt}}(p_i)$; a memory bank $\mathcal{M}_i$ stores the past $K$ embeddings $E_i$ preceding frame $x_i$. A memory attention module $\mathcal{A}$ integrates $F_i$, $M_i$, and $Q_i$ to generate an enhanced representation. Finally, a mask decoder $\mathcal{D}$ takes this representation and predicts the segmentation mask $y_i$. We can simplify the above process as follows:

$$\mathcal{Y} = f_\theta(\mathcal{X}, \mathcal{P}) \tag{1}$$

Following [42, 48], we assume that attackers are able to obtain the open-source SAM2 and can collect publicly available datasets from the Internet to make adversarial examples. The attackers' objective is to craft an adversarial perturbation $\delta$ for each frame that prevents SAM2 from accurately segmenting target objects across different prompts, *i.e.*, a cross-prompt universal adversarial attack. Furthermore, $\delta$ should be sufficiently small and constrained by the $l_p$ norm of the predefined perturbation magnitude $\epsilon$. Next, we formally define this type of attack.

**Definition 2.1** (*Cross-prompt universal adversarial attacks for SAM2*). *For an input sequence of frames $\mathcal{X}$, we generate a UAP $\delta$ for each frame $x_i \in \mathcal{X}$ to shift its predicted mask away from its ground truth $y_i$ under different prompt $\mathcal{P}$. This problem can be formulated as:*

$$\min_\delta \mathbb{E}_{x_i \sim \mathcal{X}} \left[ \forall p_i \in \mathcal{P}, \ IoU(f_\theta(x_i + \delta, p_i), y_i) \right], \quad s.t. \|\delta\|_p \leq \epsilon \tag{2}$$

In this paper, we evaluate UAP-SAM2 on both video and image segmentation tasks, with a primary focus on UAPs. For fair comparison with existing work, we also adapt our proposed approach into a *sample-wise form* UAP-SAM2* without modifying the loss function.

## 3 Methodology

### 3.1 Observation and Design Philosophy

Motivated by the significant performance gap between SAM and SAM2 under existing attacks shown in Fig. 1, we investigate the underlying causes by examining their architectural differences. We then hope to design an effective UAP for the emerging SAM2 based on our findings.

**Observation I: First-frame attacks fail to transfer to later frames.** The first design difference lies in the prompting strategy. Unlike SAM, which provides a sample-level prompt for each frame, SAM2 offers only an initial prompt on the first frame, which is then *stored and reused for segmenting subsequent frames*. Existing video attacks [37] show that image-level perturbations can transfer to video, a natural idea is to attack only the first frame and examine the effect on later frames. We apply DarkSAM as a one-shot attack to the first frame in the YouTube dataset to evaluate its effectiveness against SAM2. After generating adversarial perturbations on the first frame, we add them to all

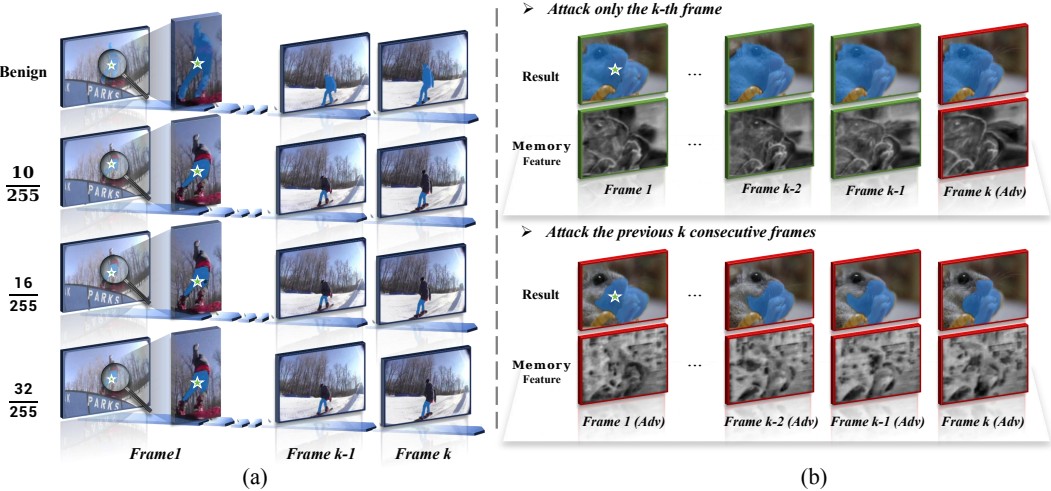

(a)

(b)

Figure 2: Visualization results of robustness analysis experiments highlighting the architectural differences between SAM2 and SAM. The pentagram denotes the point prompt on the first frame.

subsequent frames and assess the attack's impact. We gradually increase the perturbation budget from $10/255$ to $32/255$ to investigate the impact of attack strength. As depicted in Fig. 2 (a), even at the highest budget of $32/255$, DarkSAM still fails to significantly degrade segmentation performance. This may be attributed to *directional guidance from the prompt* and the *enhanced robustness of SAM2*, likely due to its advanced architecture and diverse training data. Moreover, perturbations that fail to disrupt the first frame typically cannot be effectively transferred to subsequent frames.

> **Insight I.** *Although attacking the first frame can somewhat mislead SAM2 across the video sequence, its effectiveness is limited. This motivates us to investigate other design differences to improve attack efficacy.*

**Observation II: Joint modeling of past and current frames hinders frame-specific attacks.** According to [28], besides using a fixed prompt from the first frame, SAM2 maintains a memory bank that stores semantic features from the past $k$ frames. A memory attention module integrates these features to guide the segmentation of the current frame. We refer to this use of both historical context and current-frame features as a dual-guidance mechanism.

To evaluate its impact, we inject adversarial noise into a randomly selected middle frame from the YouTube dataset. We then visualize the features extracted by the image encoder from preceding frames and stored in the memory bank. As illustrated in the 1st and 2nd rows of Fig. 2 (b), attacking a single frame alone does not significantly degrade SAM2's segmentation accuracy due to semantic entanglement across consecutive frames. However, the 3rd and 4th rows show that perturbing features from past frames to disrupt the memory bank noticeably impairs segmentation performance on the current frame. To probe the vulnerability of this dual-guidance mechanism, we examine two complementary perspectives: *semantic misalignment in the current frame* and *semantic discontinuity across adjacent frames*.

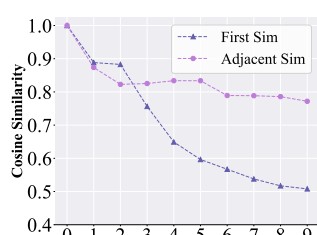

Figure 3: Avalanche effect

We segment each frame by thresholding the output masks into foreground and background. We focus on forcing SAM2 to misinterpret foreground objects as background while simultaneously enhancing background saliency to confuse the current-frame features. Fig. 2 already suggests that single-frame attacks are insufficient due to the memory module's influence. We therefore extend the attack by injecting a UAP across consecutive frames to amplify inter-frame semantic gaps. From Fig. 3, this results in a progressive decline in similarity between the current frame and both adjacent frames and the first frame. We refer to this finding as the *avalanche effect* phenomenon.

> **Insight II.** *Attacking only a single frame is ineffective in the presence of memory-based guidance. Instead, disrupting both the current semantics and temporal consistency across frames creates a stronger mismatch between guidance and segmentation, undermining SAM2's understanding of video content.*

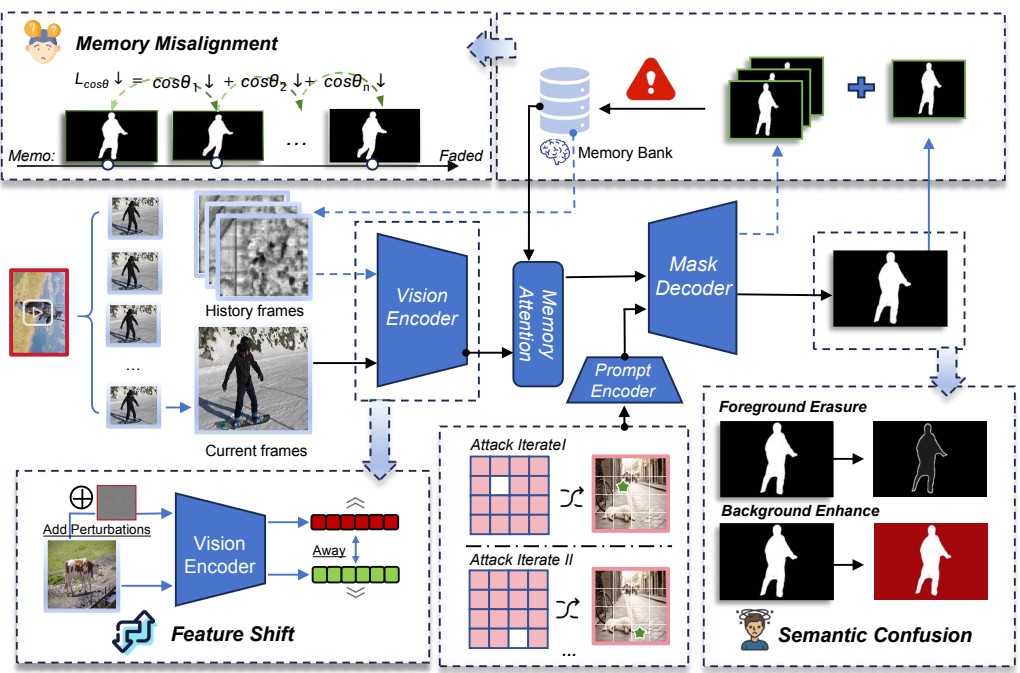

Figure 4: The framework of UAP-SAM2

## 3.2 UAP-SAM2: A Complete Illustration

Based on the design philosophy outlined in Sec. 3.1, we construct our attack from two perspectives: (i) semantic distortion within the current frame and (ii) semantic discontinuity across consecutive frames. For the current frame, we enhance attack effectiveness by jointly introducing semantic confusion and feature shift. To reduce our method's reliance on a specific prompt, we design a target-scanning strategy that selects random prompts during optimization. Specifically, we divide each video frame evenly into $m$ regions and randomly generate one prompt per region. Moreover, our optimization primarily targets the output features of the image encoder, whose input is only the image.

In this section, we present UAP-SAM2, a novel cross-prompt universal adversarial attack driven by dual semantic deviation against SAM2. As depicted the in Fig. 4, the UAP-SAM2 pipeline implements a memory-misalignment attack $\mathcal{J}_{ma}$ to disrupt temporal guidance, a feature shift attack $\mathcal{J}_{fa}$ to distort local representations, and a semantic confusion attack $\mathcal{J}_{sa}$ to confuse object-level semantics. The overall optimization objective of UAP-SAM2 is as follow:

$$\mathcal{J}_{total} = \mathcal{J}_{sa} + \mathcal{J}_{fa} + \mathcal{J}_{ma} \tag{3}$$

**Semantic confusion attack.** We apply a binary mask $m_+$ to separate the object from the background in each frame. Similar to prior attacks [48] on SAM, we aim to mislead the model by optimizing the foreground region to resemble the background. Meanwhile, we further shift foreground pixels near the decision boundary toward the background class, while reinforcing pixels originally identified as background to preserve their classification. By adding the UAP to the target frame $x_i$, we obtain the adversarial frame $\tilde{x}_i$. This objective can be formalized as:

$$\mathcal{J}_{sa} = \frac{1}{N} \sum_{i=1}^{N} \left\{ [(f_\theta(\tilde{x}_i, \mathcal{P}) \cdot m_+ - y_-)]^2 + [((1 - f_\theta(\tilde{x}_i, \mathcal{P})) \cdot m_- - y_-)]^2 \right\} \tag{4}$$

where $y_-$ is a mask that matches the frame shape, containing threshold values (*e.g.*$-1$) in regions corresponding to the target objects, and $0$ elsewhere. $m_-$ represents a binary mask for the background regions in each frame, which is the opposite of the mask that highlights the foreground.

To further effectively confuse the foreground and background, we use the *binary cross-entropy* (BCE) loss function, treating logits close to zero as having low confidence in the model's classification.

In contrast, the larger the absolute value of the logits (whether positive or negative), the higher the model's confidence in its prediction. To enhance the attack's effectiveness, we increase the overall confidence of pixel positions, naturally strengthening the updates on pixels near the decision boundary (i.e., logits close to 0), thereby pushing their logits toward the background and achieving a stronger confusion effect. Accordingly, we define the semantic confusion attack $\mathcal{J}_{sa}$ as follow:

$$\mathcal{J}_{sa} = \frac{1}{N} \sum_{i=1}^{N} \left[ \text{BCE}\left(f_\theta(\tilde{x}_i, \mathcal{P}) \cdot m_+, y_-\right) + \text{BCE}\left((1 - f_\theta(\tilde{x}_i, \mathcal{P})) \cdot m_-, y_-\right) \right] \tag{5}$$

**Feature shift attack.** We optimize the UAP to minimize the similarity between the features of the perturbed and benign frames extracted by SAM2's image encoder. We formalize this as:

$$\mathcal{J}_{fa} = -\frac{1}{N} \sum_{i=1}^{N} \cos\left(\mathcal{E}_{\text{img}}(\tilde{x}_i)\mathcal{E}_{\text{img}}(x_i)\right) \tag{6}$$

To further increase the feature discrepancy between adversarial and benign frames, we adopt a contrastive learning [2] approach. We first apply $\rho$ times random augmentations $\mathcal{T}(\cdot)$ to the target frame and aggregate their features into a prototype through $e_i = \frac{1}{\rho} \sum_{j=1}^{\rho} \mathcal{E}_{\text{img}}(\mathcal{T}(x_i))$. We then treat the adversarial frame $\tilde{x}_i$ and the prototype $e_i$ of the original frame as a negative pair, while randomly sampling frames from other videos as positive pairs. By maximizing the distance between the $\tilde{x}_i$ and $e_i$ and minimizing the distance among positive samples, we effectively drive the adversarial features away from their original semantics. Therefore, we can obtain $\mathcal{J}_{fa}$:

$$\mathcal{J}_{fa} = -\frac{1}{N} \sum_{i=1}^{N} \log \frac{\exp\left(\cos\left(\mathcal{E}_{\text{img}}(\tilde{x}_i), e_i\right)/\tau\right)}{\sum_{k=1}^{N} \mathbf{1}_{k \neq i} \exp\left(\cos\left(\mathcal{E}_{\text{img}}(\tilde{x}_i), \mathcal{E}_{\text{img}}(x_k)\right)/\tau\right)} \tag{7}$$

where $\mathbf{1}_{k \neq i}$ is an indicator function and $\tau$ denotes a temperature parameter.

**Memory misalignment attack.** Starting from the second frame, we disrupt the memory bank in SAM2 by maximizing the feature discrepancy between consecutive adversarial frames. By progressively increasing the semantic difference between the current adversarial frame and the previous one, we induce the avalanche effect illustrated in Fig. 3. This process is formulated as:

$$\mathcal{J}_{ma} = -\frac{1}{N} \sum_{i=1}^{N} \cos\left(\mathcal{E}_{\text{img}}(\tilde{x}_{i+1}), \mathcal{E}_{\text{img}}(\tilde{x}_i)\right) \tag{8}$$

## 4  Experiments

### 4.1  Experimental Setup

**Datasets and models.** We evaluate our attack on there public video segmentation datasets: YouTube-VOS2018 (YouTube) [8], DAVIS 2017 (DAVIS) [26], and MOSE [6] for video segmentation tasks. To further investigate the performance of UAP-SAM2 on image segmentation tasks, we construct corresponding image-based datasets by randomly sampling frames from the original video datasets, denoted as YouTube*, DAVIS*, and MOSE*. We resize all frames in the videos to a uniform size of 3×1024×1024. We use pre-trained SAM2-T, SAM2-S, and SAM2.1-T from the official repository as the target models. To further validate the transferability, we will conduct evaluations on Sam2long [7], which enhances the capabilities of SAM2 in long-video tasks. More details are in Appendix-B.

**Attack settings.** We set the perturbation bound $\epsilon$ of the universal adversarial attack UAP-SAM2 to $10/255$, and that of the sample-wise variant attack UAP-SAM2* to $8/255$, using a batch size of 1 and training for 10 epochs. We use a fixed random seed of 30 for all experiments to ensure reproducibility. We use point prompts for the default evaluation.

**Evaluation metrics.** Following [42, 48], we use the *mean Intersection over Union* (mIoU) metric to evaluate the effectiveness of UAP-SAM2. mIoU is a widely used metric in segmentation tasks [4, 24, 28] that measures the average overlap between the predicted and ground truth segmentation masks. A lower mIoU value indicates better attack performance.

Table 1: mIoU (%) results of adversarial examples under different settings

| Setting | | Video segmentation | | | | | | Image segmentation | | | | | |
|---|---|---|---|---|---|---|---|---|---|---|---|---|---|
| | | Point | | | Box | | | Point | | | Box | | |
| | | $\mathcal{M}_1$ | $\mathcal{M}_2$ | $\mathcal{M}_3$ | $\mathcal{M}_1$ | $\mathcal{M}_2$ | $\mathcal{M}_3$ | $\mathcal{M}_1$ | $\mathcal{M}_2$ | $\mathcal{M}_3$ | $\mathcal{M}_1$ | $\mathcal{M}_2$ | $\mathcal{M}_3$ |
| UAP | $\mathcal{D}_1$ | 37.03 | 36.26 | 42.47 | 45.17 | 28.39 | 41.06 | 27.54 | 28.36 | 21.89 | 52.07 | 27.19 | 42.36 |
| | $\mathcal{D}_2$ | 42.47 | 40.85 | 52.08 | 49.03 | 38.28 | 48.47 | 48.45 | 47.20 | 46.47 | 44.51 | 37.32 | 45.83 |
| | $\mathcal{D}_3$ | 33.67 | 37.03 | 52.46 | 49.13 | 34.38 | 49.66 | 50.13 | 52.22 | 49.16 | 61.63 | 40.28 | 52.03 |
| | AVG | 37.72 | 38.05 | 49.00 | 47.78 | 33.68 | 46.40 | 42.04 | 42.59 | 39.17 | 52.74 | 34.93 | 46.74 |
| Sample-wise | $\mathcal{D}_1$ | 45.39 | 27.72 | 39.02 | 57.72 | 55.18 | 61.57 | 33.42 | 28.49 | 34.19 | 32.42 | 29.28 | 37.93 |
| | $\mathcal{D}_2$ | 41.64 | 23.57 | 43.24 | 50.26 | 43.37 | 51.73 | 36.92 | 31.44 | 39.21 | 35.62 | 30.81 | 37.65 |
| | $\mathcal{D}_3$ | 46.60 | 35.91 | 49.17 | 60.69 | 52.88 | 60.89 | 45.81 | 38.42 | 42.51 | 48.13 | 43.44 | 50.71 |
| | AVG | 44.54 | 29.07 | 43.81 | 56.22 | 50.48 | 58.06 | 38.72 | 32.78 | 38.64 | 38.72 | 34.51 | 42.10 |

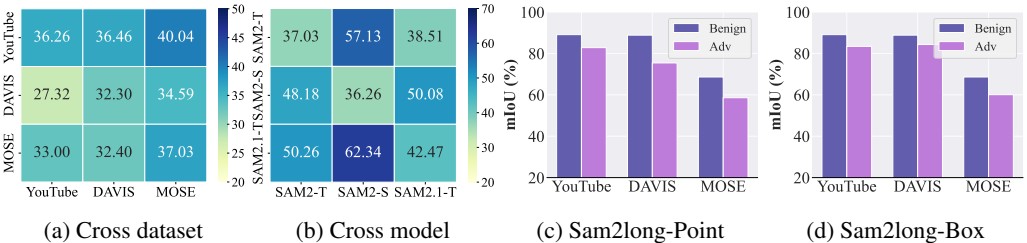

| (a) Cross dataset | (b) Cross model | (c) Sam2long-Point | (d) Sam2long-Box |

Figure 5: Transferability study. (a) shows the results of cross-dataset transferability studies, while (b) - (d) present the results of cross-model transferability studies.

**Platform.** Experimental hardware details. We conduct experiments on a machine with two NVIDIA A100-SXM4 GPUs, two Intel(R) Xeon(R) Gold 6132 CPUs and 314GB RAM.

## 4.2 Attack Performance

To comprehensively evaluate the effectiveness of UAP-SAM2, we conduct experiments on six datasets (YouTube, DAVIS, MOSE, YouTube*, DAVIS*, and MOSE*) covering both video and image segmentation tasks. We evaluate three model variants: SAM2-T, SAM2-S, and SAM2.1-T. For clarity, we denote the datasets (including their corresponding variant datasets) as $\mathcal{D}_1$–$\mathcal{D}_3$ and the models as $\mathcal{M}_1$–$\mathcal{M}_3$ throughout the paper. We evaluate both the sample-wise and universal variants of our attack under 72 different settings. For each setting, we generate adversarial examples using both point and box prompts, and report performance under point-prompt evaluation. As a reference, Fig. 6 shows SAM2's segmentation accuracy on benign samples. Across six datasets for both image and video segmentation tasks, SAM2 achieves an average mIoU above 76%, demonstrating strong segmentation capability and generalization. Tab. 1 shows that adversarial

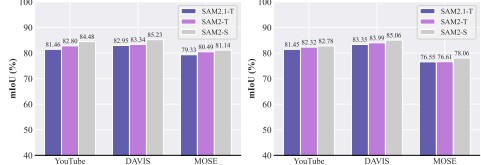

Figure 6: Left: Video, Right: Image

examples generated by UAP-SAM2 consistently and significantly degrade SAM2's performance with cross-prompt transferability. Notably, on the DAVIS dataset with point prompts, UAP-SAM2 and its variant UAP-SAM2* reduce SAM2's mIoU by over 45.79% and 54.77%, respectively. As presented in Tab. 1, regardless of whether point or box prompts are used, our method consistently achieves stronger attack performance on video segmentation than on image segmentation. This further validates its effectiveness in disrupting semantic consistency across consecutive frames.

We further evaluate the transferability of our approach across different datasets and models. Fig. 5 (a) and Fig. 5 (b) report the performance of UAP-SAM2 under transfer settings, where each row corresponds to adversarial examples generated from the same source. The results demonstrate strong transferability across datasets and models. Additionally, Fig. 5 (c) and Fig. 5 (d) show the attack performance of UAPs crafted on SAM2-T and transferred to Sam2long [7] under point and box prompts, confirming the effectiveness of our method against SAM2 variants as well.

Table 2: (UAP) The mIoU(%) results of the comparison study under different settings. Bold indicates the best results. Since UAD does not use prompts during the optimization process, the results are identical under both box-prompt and point-prompt settings.

| Method | Point | | | | | | Box | | | | | |
| | Video | | | Image | | | Video | | | Image | | |
| | $\mathcal{D}_1$ | $\mathcal{D}_2$ | $\mathcal{D}_3$ | $\mathcal{D}_1$ | $\mathcal{D}_2$ | $\mathcal{D}_3$ | $\mathcal{D}_1$ | $\mathcal{D}_2$ | $\mathcal{D}_3$ | $\mathcal{D}_1$ | $\mathcal{D}_2$ | $\mathcal{D}_3$ |
|---|---|---|---|---|---|---|---|---|---|---|---|---|
| UAPGD [5] | 42.59 | 53.60 | 50.80 | 54.42 | 50.11 | 61.76 | 65.01 | 56.97 | 62.33 | 64.79 | 50.11 | 63.42 |
| VOSPGD [13] | 60.91 | 54.24 | 63.47 | 50.05 | 48.56 | 53.52 | 63.06 | 56.20 | 65.15 | 61.88 | 51.63 | 56.21 |
| SegPGD [10] | 43.24 | 52.34 | 58.89 | 56.22 | 49.96 | 61.02 | 63.43 | 58.54 | 65.26 | 64.73 | 52.64 | 62.71 |
| AttackSAM [42] | 64.35 | 62.31 | 63.05 | 64.18 | 55.53 | 63.92 | 60.88 | 58.80 | 57.12 | 63.72 | 51.75 | 62.66 |
| S-RA [29] | 61.18 | 56.20 | 60.25 | 63.53 | 51.04 | 54.38 | 62.45 | 57.89 | 60.82 | 62.01 | 50.05 | **53.57** |
| UAD [22] | 49.39 | 53.66 | 53.51 | 56.12 | 51.80 | 61.87 | 49.39 | 53.66 | 53.51 | 56.12 | 51.80 | 61.87 |
| DarkSAM [48] | 67.51 | 57.00 | 51.96 | 64.38 | 52.99 | 64.38 | 66.20 | 58.91 | 62.76 | 65.58 | 53.76 | 65.03 |
| Ours | **37.03** | **42.47** | **33.67** | **27.54** | **48.45** | **50.13** | **45.17** | **49.03** | **49.13** | **52.07** | **44.51** | 61.63 |

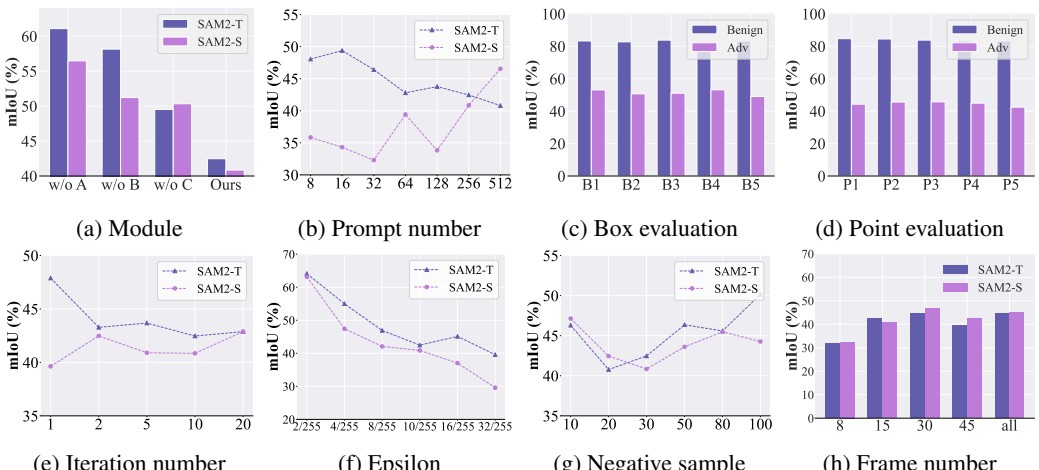

(a) Module  (b) Prompt number  (c) Box evaluation  (d) Point evaluation

(e) Iteration number  (f) Epsilon  (g) Negative sample  (h) Frame number

Figure 7: Ablation results on the effect of different factors on the attack performance of UAP-SAM2

## 4.3 Comparison Study

Given the absence of adversarial attacks tailored for SAM2, we evaluate our method comprehensively by comparing UAP-SAM2 against the latest adversarial attacks targeting SAM, such as Attack-SAM [42], S-RA [29], UAD [22], and DarkSAM [48]. We further compare UAP-SAM2 against representative adversarial attack methods originally designed for classification, image segmentation, and video segmentation tasks, including UAPGD [5], SegPGD [10], and VOSPGD [13]. To ensure a fair comparison, we adapt all baseline methods into a universal adversarial attack framework and apply the same optimization settings as used in UAP-SAM2. To evaluate the cross-prompt generalization ability of these methods, we uniformly adopt random prompts, *i.e.*, the prompts used during training and testing are different, for method optimization to generate adversarial examples. We choose SAM2-T as the target model and evaluate the performance of all methods on both image and video segmentation tasks across six datasets. As depicted in Tab. 2, UAP-SAM2 outperforms all existing attacks on video segmentation across three datasets. For image segmentation, our method also surpasses most baselines. The above results can be attributed to the tailored design of video features in our method.

## 4.4 Ablation Study

In this section, we investigate the different factors on the attack performance of UAP-SAM2. We use SAM2-T and SAM2-S as the target models, with DAVIS as the dataset.

**The effect of the module.** We perform ablation studies to assess the contribution of individual components to the attack effectiveness of UAP-SAM2. For clarity, we denote $\mathcal{L}_{sa}$, $\mathcal{L}_{fa}$, and $\mathcal{L}_{ma}$

as A, B, and C, respectively. As observed in Fig. 7 (a), none of the ablated variants surpass the full model, highlighting the importance of each module in achieving optimal attack performance.

**The effect of prompt numbers.** We investigate how the prompt number $m$ of segmented regions in the proposed target-scanning strategy influences the attack performance of UAP-SAM2. We vary the region count from 8 to 512 and report the results in Fig. 7 (b). The attack achieves optimal performance under both settings when $m = 256$, which we adopt as the default configuration.

**The effect of evaluation modes.** We study how different evaluation prompt settings affect the attack performance of UAP-SAM2. Specifically, we evaluate the perturbation generated on SAM2-T using five randomly sampled box prompts (B1 – B5) and five point prompts (P1 – P5). As depicted in Fig. 7 (c) - (d), UAP-SAM2 consistently maintains strong performance across different prompt configurations, demonstrating the robustness of our method.

**The effect of iteration numbers.** We investigate the effect of the iteration numbers on attack performance of UAP-SAM2. We conduct experiments with varying numbers of iterations, ranging from 1 to 20. The results shown in Fig. 7 (e) indicate that the attack performance stabilizes after the number of iterations reaches 10. Therefore, we set it as the default configuration for our experiments.

**The effect of $\epsilon$.** We evaluate UAP-SAM2's performance with $\epsilon$ from $2/255$ to $32/255$ in Fig. 7 (f). With the increase in $\epsilon$, there is a corresponding enhancement in attack performance. Notably, even at the $4/255$ setting, our method still maintains high attack efficacy, with an average mIoU decrease of over $33.08\%$.

**The effect of negative sample numbers.** We explore the effect of varying the number of negative samples from 10 to 100 on the performance of UAP-SAM2 in Fig. 7 (g). Considering both computational efficiency and attack effectiveness, we set 30 as our default testing.

**The effect of testing frame numbers.** We study the effect of the number of selected frames per video on the performance of UAP-SAM2. As shown in Fig. 7 (h), the results with 15 frames are comparable to those using all frames. Therefore, for efficiency considerations, we set 15 frames as the default configuration for our experiments.

## 5  Defense Study

Since there is no dedicated adversarial defense tailored for SAM2, we explore the robustness of UAP-SAM2 through two common defense strategies: *model pruning* [50] and *data pre-processing* [27].

**Model pruning** is a widely used compression technique that removes redundant parameters to simplify network complexity, thereby potentially reducing sensitivity to perturbations. We evaluate the attack performance under various pruning ratios ranging from 0 to 0.9 on DAVIS. As shown in Fig. 8 (a) - (b), the mIoU of benign samples consistently decreases as the pruning ratio increases, whereas that of adversarial examples remains relatively stable. Notably, even when the pruning ratio reaches $0.4$, the performance of benign samples degrades significantly, while the mIoU of adversarial examples remains largely unaffected. These results suggest that model pruning offers limited robustness against UAP-SAM2.

**Data pre-processing** suppresses the impact of adversarial noise by introducing distortions such as occlusion or blur into the image. We apply spatter (sp_) and saturate (sa_) corruption at severity levels from 0 to 5 to assess the effectiveness of this strategy on DAVIS. As observed in Fig. 8 (c) - (d), increasing the corruption strength leads to a consistent drop in benign sample mIoU, while adversarial mIoU remains largely unaffected. These findings suggest that our method remains effective even when facing pre-processing defenses based on input corruption.

## 6  Related Works

### 6.1  Segment Anything Models

*Segment Anything Model* (SAM) [17] has achieved remarkable success in image segmentation due to its strong generalization ability. SAM2 [28], the latest improved version, is applied to both image and video segmentation tasks through the memory mechanism, extending SAM to general-purpose video segmentation. The user only needs to provide a prompt on the first frame, and SAM2 can

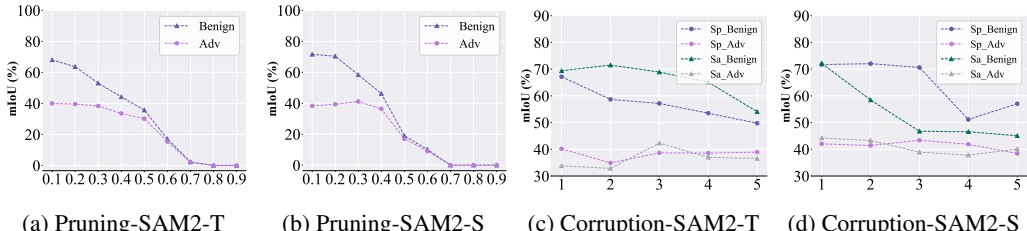

|  |  |  |  |
|---|---|---|---|
| (a) Pruning-SAM2-T | (b) Pruning-SAM2-S | (c) Corruption-SAM2-T | (d) Corruption-SAM2-S |

Figure 8: The results (%) of the defense study. (a) – (b) show the experimental results of parameter pruning, (c) – (d) present the results of input pre-processing.

then perform real-time object localization and segmentation in subsequent frames. Building on its strong generalization, recent studies [1, 3, 7, 15, 19, 32, 39, 49] develop task-specific variants of SAM to better address various downstream applications. SAM2 has been rapidly applied to various downstream tasks, such as medical video segmentation [49], 3D segmentation [11], and camouflaged object detection [32].

## 6.2 Adversarial Attacks on SAM

Recent studies [12, 20, 21, 29, 38, 42, 48] reveal that the SAM is vulnerable to adversarial examples [9, 23, 25, 47, 18, 31, 44], which are crafted by adding imperceptible perturbations to induce incorrect predictions. Existing adversarial attacks on SAM fall into two categories: sample-wise, which tailor perturbations to each input, and universal, which create a single perturbation that works across many images. Attack-SAM [42] is the first to adopt PGD [23] to manipulate the predicted masks of image–prompt pairs. UAD [22] extends this direction by simulating spatial deformations to optimize adversarial noise that disrupts the feature representations of the image encoder, enabling prompt-free attacks. In parallel, DarkSAM [48] introduces the first universal adversarial attack against SAM. It designs a hybrid spatial-frequency framework that prevents objects in the image from being segmented and proposes a shadow target strategy to improve cross-prompt transferability. Other studies [20, 29] focus on localized attacks that deceive SAM into failing to segment specific objects within an image. Despite their effectiveness on SAM, these methods cannot be directly applied to SAM2 due to the modality gap between images and videos, and the architectural novelties of SAM2.

## 7 Conclusions, Limitations, and Broader Impact

In this paper, we investigate the performance gap of existing attacks against SAM and SAM2, and attribute it to two key challenges: directional guidance from the prompt and semantic entanglement across consecutive frames. To this end, we propose UAP-SAM2, the first cross-prompt universal adversarial attack driven by dual semantic deviation against SAM2. We design a target-scanning strategy and directly perturb the output features of the image encoder to enhance the cross-prompt transferability. To further boost the attack effectiveness, we jointly exploit semantic confusion and feature deviation. We conduct extensive experiments on six datasets across two segmentation tasks to demonstrate the effectiveness of the proposed method for SAM2.

While our work focuses on prompt-based video segmentation models, a potential limitation is that UAP-SAM2 may not directly generalize to traditional segmentation models, as their outputs are not label-free masks. Although SAM-based models are gaining popularity, it remains unclear how well adversarial examples crafted for SAM2 transfer to other segmentation frameworks. As these models are increasingly adopted in safety-critical applications such as autonomous driving and medical imaging, understanding their vulnerabilities is a crucial direction for future research.

## Acknowledgements

This work is sponsored by the Major Program (JD) of Hubei Province (No.2023BAA024) and National Natural Science Foundation of China under Grants Nos.62372196 and 62202186. Yufei Song is the is the corresponding author.

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

# A  Contents

# B  Experimental Setting

In this section, we provide details of our experimental settings. For video segmentation, we randomly select 100 videos and sample 15 consecutive frames from each for evaluation. For image segmentation, we randomly choose 50 videos and uniformly sample a total of 500 frames.

## B.1  Datasets

- **DAVIS 2017:** DAVIS 2017 [26] is a standard dataset widely used for video target segmentation tasks. Its training set contains 60 videos and the test set contains 30 videos. Each video provides pixel-level target (human, animal, object) segmentation annotations, that is, each frame gives the precise boundary of the target. It is specially designed for target segmentation and tracking tasks in videos, especially suitable for multi-target tracking and segmentation research.

- **YouTube-VOS2018:** YOUTUBE-VOS2018 [8] is a large-scale dataset designed for video object segmentation tasks based on video content on the YouTube platform, especially for accurate object segmentation in long video sequences. This dataset provides large-scale, densely annotated video sequences. The training set contains 3,883 videos involving 40 different object categories, and the test set contains 1,474 videos involving 20 different object categories.

- **MOSE:** MOSE [6] is a large-scale dataset designed for video object segmentation tasks based on video content on the YouTube platform, especially for accurate object segmentation in long video sequences. This dataset provides large-scale, densely annotated video sequences. The training set contains 3,883 videos involving 40 different object categories, and the test set contains 1,474 videos involving 20 different object categories.

## B.2  Evaluation Metrics

We choose *Mean Intersection over Union* (mIoU) as the metric to evaluate segmentation accuracy. mIoU is a commonly used evaluation method in semantic segmentation tasks to measure the model's performance across different categories. It is calculated by determining the *Intersection over Union* (IoU) for each category and averaging the IoUs across all categories to obtain the final evaluation result. Specifically, IoU is the ratio of the intersection area between the predicted region and the ground truth region to the union of both areas. The formula for calculating IoU for each category is:

$$\text{IoU} = \frac{\text{Predicted Region} \cap \text{Ground Truth Region}}{\text{Predicted Region} \cup \text{Ground Truth Region}}$$

mIoU is the average of the IoUs for all categories, reflecting the model's overall performance in the segmentation task. A higher mIoU indicates better segmentation performance across categories, particularly in tasks with class imbalance or fine-grained segmentation.

# C  Supplementary Comparison Study

Consistent with Sec. 4.3, we compare UAP-SAM2* against a range of SOTA adversarial attacks, including Attack-SAM [42], S-RA [29], UAD [22], DarkSAM [48], PGD [23], SegPGD [10], and VOSPGD [13]. To ensure a fair comparison, all baseline methods are adapted to a sample-wise adversarial attack framework and optimized under the same settings as UAP-SAM2*. To evaluate the cross-prompt generalization ability of these methods, we uniformly adopt random prompts, *i.e.*, the prompts used during training and testing are different, for method optimization to generate adversarial

Table A1: (Sample-wise) The mIoU (%) results of the comparison study under different settings. Bold indicates the best results. Since UAD does not use prompts during the optimization process, the results are identical under both box-prompt and point-prompt settings.

| Method | Point | | | | | | Box | | | | | |
| | Video | | | Image | | | Video | | | Image | | |
| | $\mathcal{D}_1$ | $\mathcal{D}_2$ | $\mathcal{D}_3$ | $\mathcal{D}_1$ | $\mathcal{D}_2$ | $\mathcal{D}_3$ | $\mathcal{D}_1$ | $\mathcal{D}_2$ | $\mathcal{D}_3$ | $\mathcal{D}_1$ | $\mathcal{D}_2$ | $\mathcal{D}_3$ |
|---|---|---|---|---|---|---|---|---|---|---|---|---|
| UAPGD [5] | 56.61 | 52.59 | 54.74 | 53.12 | 49.84 | 65.03 | 71.75 | 58.67 | 60.63 | 61.90 | 54.44 | 64.92 |
| VOSPGD [13] | 57.66 | 52.76 | 60.48 | 53.59 | 49.99 | 65.31 | 67.99 | 59.69 | 67.60 | 62.33 | 54.43 | 64.57 |
| SegPGD [10] | 53.27 | 51.73 | 51.92 | 51.91 | 49.58 | 63.11 | 67.78 | 55.95 | 67.70 | 62.27 | 55.44 | 61.74 |
| AttackSAM [42] | 50.00 | 45.76 | 50.99 | 39.09 | 39.54 | 56.77 | **52.80** | 53.14 | 63.61 | 52.78 | 47.61 | 62.88 |
| S-RA [29] | 47.86 | 54.90 | 49.33 | 35.29 | 52.61 | 53.30 | 57.98 | 54.37 | 56.21 | 65.61 | 60.73 | 65.25 |
| UAD [22] | 55.51 | 60.20 | 47.89 | 45.41 | 48.69 | 48.74 | 55.51 | 60.20 | **47.89** | 45.41 | 48.69 | 48.74 |
| DarkSAM [48] | 61.96 | 49.53 | 58.37 | 58.25 | 50.47 | 61.57 | 60.46 | 56.55 | 65.51 | 59.51 | 52.57 | 63.79 |
| Ours | **45.39** | **41.64** | **46.60** | **33.42** | **36.92** | **45.81** | 57.72 | **50.26** | 60.69 | **32.42** | **35.62** | **48.13** |

examples. We adopt SAM2-T as the target model and evaluate all methods on both image and video segmentation tasks across six datasets. As shown in Tab. A1, UAP-SAM2* generally outperforms all existing attack methods across image and video segmentation tasks on three datasets, with only one exception. Visual comparisons under the UAP and sample-wise adversarial attack frameworks are provided in Fig. A3 and Fig. A4, respectively.

## D Multi-point Evaluation Study

We conduct an in-depth investigation into the impact of the multi-prompt evaluation settings on the effectiveness of our proposed attack method. Specifically, we select SAM2-T as the target model and examine how varying the number of input prompts influences segmentation performance on adversarial examples. As illustrated in Fig. A2, we present qualitative visualizations of SAM2's segmentation outputs under different numbers of prompt points. Our observations indicate that increasing the number of prompts provides the model with more spatial information about the target objects, which can slightly mitigate the impact of the adversarial perturbations. Nevertheless, UAP-SAM2 continues to exhibit strong attack performance, consistently disrupting SAM2's ability to generate coherent and semantically meaningful segmentations, even under dense prompting conditions. These results highlight the robustness and general effectiveness of our method across varied prompt configurations.

## E Stability Analysis

Considering the potential influence of random seed settings on the selection of training and testing images, we conduct a detailed analysis of how different random seeds affect the performance of UAP-SAM2. While a default random seed of 30 is adopted in all our main experiments to ensure consistency, we further explore the robustness of our method by evaluating it under multiple random seed settings. Specifically, we select five random seeds and perform universal adversarial attacks using SAM2-T as the target model on three benchmark datasets: DAVIS, YouTube, and MOSE. As illustrated in Fig. A1, the error bars reflect the variance in attack performance across different seeds. The consistently small fluctuations across datasets confirm that UAP-SAM2 delivers stable and reliable results, demonstrating strong robustness to variations in seed initialization.

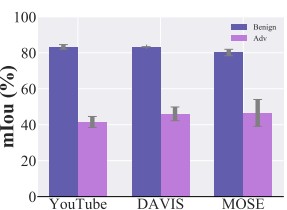

Figure A1: Stability analysis

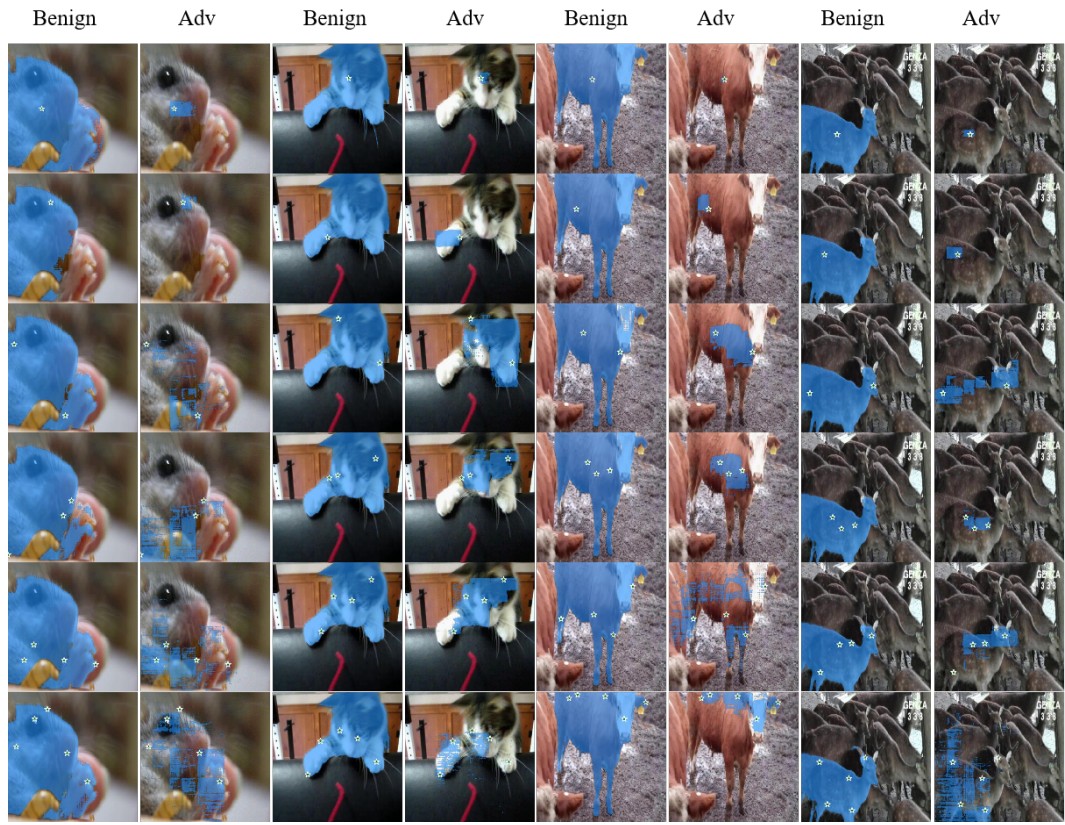

Figure A2: Visualization of SAM2 segmentation results on adversarial examples generated by UAP-SAM2 under the multipoint evaluation mode

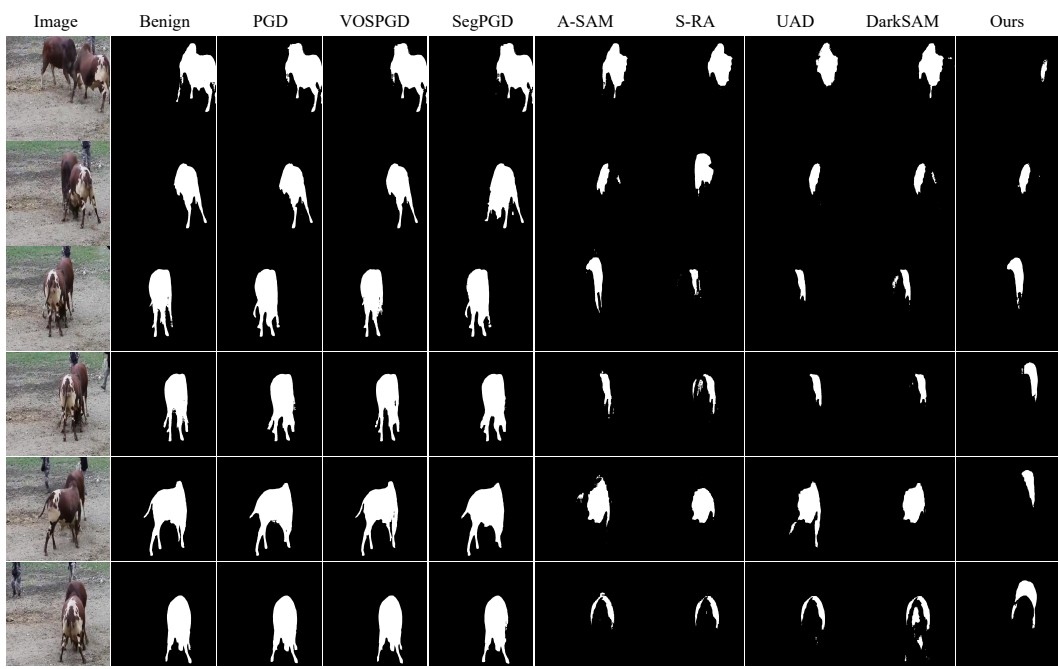

Figure A3: (UAP) Visualization of the comparison study results

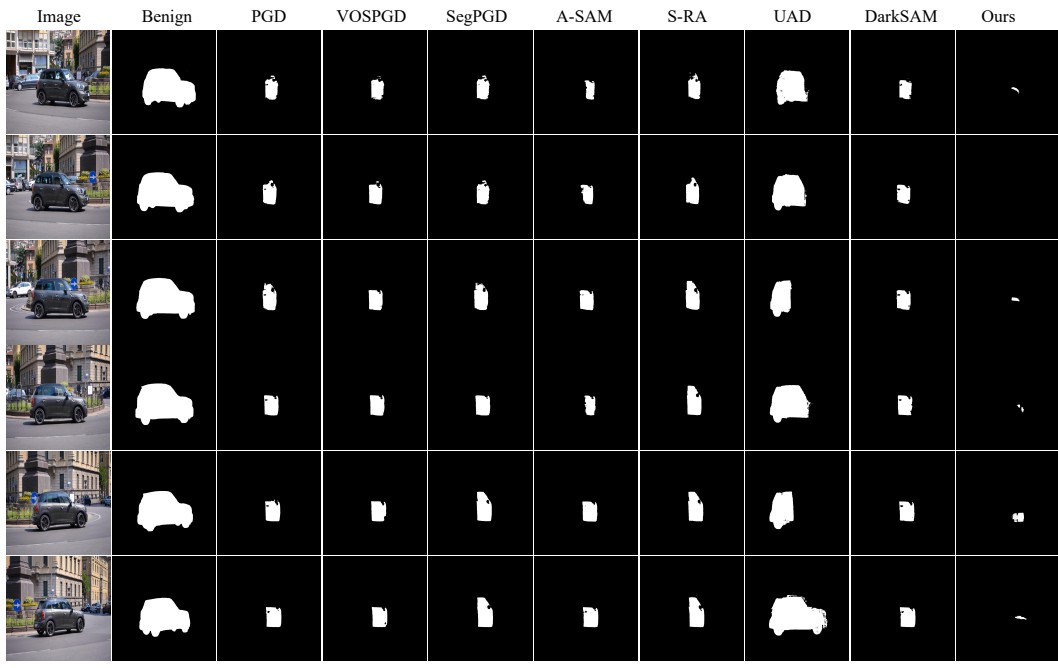

Figure A4: (Sample-wise) Visualization of the comparison results

