# OpenReview forum: "Vanish into Thin Air: Cross-prompt Universal Adversarial Attacks for SAM2"
_NeurIPS.cc/2025/Conference — NeurIPS 2025 spotlight_

### Official Review · Reviewer_N6mr · 2025-06-13

**Clarity:** 3
**Significance:** 4
**Originality:** 3
**Rating:** 5
**Confidence:** 3

**Summary:**

SAM2 has received a lot of attention due to its strong generalization ability in video segmentation. However, the robustness of the model has not been explored, and it is unclear whether existing attacks on SAM are directly transferable to SAM2. The authors analyze attack transferability and identify two key challenges: the directional role of the prompt and semantic entanglement across consecutive frames. To address these challenges, they propose UAP-SAM2, a cross-prompt universal adversarial attack on SAM2. A target scanning approach is used to reduce prompt dependency, and the UAP is optimized to disrupt both the semantics in the current frame and the consistency between frames. Extensive experiments are conducted on six datasets and different models.

**Questions:**

- Why were no image segmentation datasets used for the second part of the experiment, as in other papers, e.g., DarkSAM, which are also more complex such as Cityscapes or SA-1B?
- The attack success rate is also frequently used as an evaluation metric. Since it looks purely at the pixels, this metric provides different information than the mIoU, which averages across categories.

**Ethical Concerns:**

["NO or VERY MINOR ethics concerns only"]

**Final Justification:**

After reading the other reviews and the rebuttal, I still find the paper acceptable.

**Limitations:**

yes

**Paper Formatting Concerns:**

-

**Quality:**

3

**Strengths And Weaknesses:**

Strengths:
- The paper is well written and easy to understand.
- The topic discussed is highly timely and security-relevant. The experiment in the introduction shows that attacks are not easily transferable from SAM to SAM2, which highlights the importance and relevance of the topic presented.
- Section 3.1 helps to understand problems with the attacks and the idea behind the proposed method. Really well thought out, which individual elements make up the attack.
- The results achieved are very good in comparison to SOTA. The comparison is fair and new methods were tested.
- The ablation study considers many possible variables and justifies the authors' choice of parameters.
- The experiments with defense methods also demonstrate the capabilities of the attack.
- The qualitative results in the appendix clearly show the differences between the methods.
- The appendix has a good length and has been used appropriately.
- The authors provide code for reproducibility.

Weaknesses:
- I would move the related work section to the beginning of the paper, firstly so that the initial experiments can be better understood and secondly so that the methodological added value of the attack presented here can be better understood.
- Are the observations in section 3.1 generally valid, i.e., also for other attacks and datasets?
- What is m_ in equation 4?

Remarks:
- In line 192, three datasets instead of four.
- I would add a sentence explaining the differences between the SAM2 models.

---

> ### Author Rebuttal · Authors · 2025-07-30
>
> ## Responses to Reviewer N6mr
>
> Thanks for your professional and detailed suggestions on our work! You can find the responses to your questions as follows:
>
> ------
>
> **Q1**: Concern regarding the  the related work section
>
> **A1**: Thank you for the constructive suggestion! We will move the related work section to the beginning of the paper in the revised version to improve the clarity of both the initial experiments and the methodological contributions of our attack.
>
> ------
>
> **Q2**: Concern regarding the  the observations in section 3.1
>
> **A2**:  We provide results for other attacks and datasets in Tables R1, R2, and R3, demonstrating that the observations in Section 3.1 are generally valid. Specifically:
>
> 1. For Observation I, we evaluate the performance of AttackSAM [1], S-RA [2], and DarkSAM [3] under varying perturbation (10/255, 16/255, 32/255) budgets on the YouTube and DAVIS datasets across SAM2-T. Table R1 shows that existing attacks targeting SAM, even with increased perturbation budgets, only partially degrade SAM2’s performance, consistent with our findings in the main text.
>
> 2. For Observation II, we further investigate the avalanche effect of adversarial frames generated by DarkSAM and our proposed attack on the YouTube and DAVIS datasets  across SAM2-T. Tables R2 and R3 indicate that adversarial frames with added noise significantly reduce feature consistency between adjacent frames, aligning with our conclusions in the main text.
>
>
>
> Tab.R1 Experimental analysis of existing attacks on SAM2 under varying perturbation budgets
>
> |  Method   |    | YouTube        |    |    |    | DAVIS         |    |    |
>
> |------------|-------|------------------------|-------|-------|-------|-----------------------|-------|-------|
>
> |   Setting  |    | Benign | 10/255 | 16/255 | 32/255 | Benign | 10/255 | 16/255 | 32/255 |
>
> | AttackSAM  |    | 82.8  | 66.94  | 60.86  | 57.15  | 84.48  | 56.00  | 54.56  | 56.19  |
>
> | S-RA    |    | 82.8  | 61.65  | 63.78  | 60.83  | 84.48  | 62.43  | 54.15 | 54.22  |
>
> | DarkSAM   |    | 82.8  | 52.77  | 56.78  | 55.49  | 84.48  | 52.19  | 50.52  | 51.84  |
>
>
>
> Tab.R2 An investigation of the avalanche effect on YouTube
>
> |       Frame      | 0 |   1   |   2   |   3    |  4    |   5    |   6    |  7    |   8    |   9    |
>
> |  DarkSAM _Adjacent | 1 | 0.9468 | 0.9604 | 0.9544 | 0.9627 | 0.9484 | 0.9478 | 0.9582 | 0.9410 | 0.9547 |
>
> |   DarkSAM _First  | 1 | 0.8560 | 0.8390 | 0.8675 | 0.8343 | 0.8012 | 0.7785 | 0.7880 | 0.7758 | 0.7788 |
>
> |Ours_ Adjacent| 1 | 0.9367 | 0.9148 | 0.9634 | 0.9452 | 0.9173 | 0.8215 | 0.8056 | 0.8123 | 0.7796 |
>
> |  Ours_First   | 1 | 0.7235 | 0.7065 | 0.7657 | 0.7295 | 0.6916 | 0.5574 | 0.5114 | 0.4972 | 0.4825 |
>
>
>
> Tab.R3 An investigation of the avalanche effect on DAVIS
>
> |       Frame      | 0 |   1   |   2   |   3    |  4    |   5    |   6    |  7    |   8    |   9    |
>
> |----------------------------|---|----------|----------|-----------|---------|-----------|----------|---------|----------|----------|
>
> |  DarkSAM _Adjacent | 1 | 0.9896 | 0.9758 | 0.9880 | 0.9886 | 0.9811 | 0.9819 | 0.9882 | 0.9796 | 0.9742 |
>
> |   DarkSAM _First    | 1 | 0.9162 | 0.9045 | 0.8914 | 0.8817 | 0.8773 | 0.8700 | 0.8631 | 0.8615 | 0.8602 |
>
> | Ours_ Adjacent | 1 | 0.8736 | 0.8228 | 0.8253 | 0.8338 | 0.8337 | 0.7892 | 0.7886 | 0.7858 | 0.7718 |
>
> |   Ours_SAM2_First  | 1 | 0.8884 | 0.8828 | 0.7564 | 0.6492 | 0.5964 | 0.5671 | 0.5379 | 0.5171 | 0.5080 |
>
>
>
> ------
>
> **Q3**: Concern regarding the Eq. 4
>
> **A3**: $m^-$  in Eq. 4  represents a binary mask for the background regions in each frame, which is the opposite of the $m^+$ mask that highlights the foreground. We will clarify this definition explicitly in the revised version.
>
> ------
>
> **Q4**: Concern regarding typos
>
> **A4**: Thanks for pointing this out. We will make the corresponding changes in the revised version.
>
> ------
>
> **Q5**: Concern regarding the differences between the SAM2 models
>
> **A5**: The differences between SAM2-T, SAM2-S, and SAM2.1-T lie in model scale and architecture. SAM2-T and SAM2-S are Tiny and Small variants of SAM2, respectively, while SAM2.1-T is an improved version of SAM2-T with architectural enhancements that offer better segmentation performance and efficiency.
>
> ------
>
> **Q6**: Concern regarding the image segmentation datasets
>
> **A6**: Thank you for the insightful suggestion. Since our work primarily focuses on attacking large video segmentation models, we conduct our main evaluations on video datasets. We provide additional results in Tab.R4, showing the performance of our attack on the SA-1B dataset under point prompts across SAM2-T, SAM2-S, and SAM2.1-T. These results demonstrate that our method also remains effective on standard image segmentation benchmarks.
>
>
>
> Tab.R4 The performance of our attack on the SA-1B datase
>
> |  mIoU ｜SAM2-T  | SAM2-S  |  SAM2.1-T |
>
> |-------------|--------------|------------|------------|
>
> |  Benign  |  69.82   |  71.45  |  69.47  |
>
> |  Adv    |  52.99   |  50.79  |  53.64   |
>
> ------
>
> **Q7**: Concern regarding the attack success rate metric
>
> **A7**: We agree with the reviewer’s interpretation of the attack success rate as a complementary metric. In this work, due to the varying segmentation capabilities of different SAM2 models, we follow prior SAM-related studies [1,2,3] and adopt mIoU as the primary evaluation metric, using its drop to indicate attack effectiveness. We will consider incorporating attack success rate in future work to provide a more comprehensive evaluation.
>
> Reference:
>
> [1] Attack-sam: Towards evaluating adversarial robustness of segment anything model, ArXiv 2024.
>
> [2] Practical region-level attack against segment anything models, CVPR 2024.
>
> [3] Darksam: Fooling segment anything model to segment nothing, NeurIPS 2024.

---

### Official Review · Reviewer_HAPs · 2025-06-16

**Clarity:** 3
**Significance:** 3
**Originality:** 3
**Rating:** 5
**Confidence:** 4

**Summary:**

This paper presents a cross prompt adversarial attack against SAM2 on both image and video segmentation tasks. Authors carefully studied the problem of existing adversarial perturbation on SAM2, based on which three losses aiming at attacking foreground semantics, inner frame feature alignment and inter frame feature alignment are proposed. Experiments on six image segmentation tasks are presented to demonstrate the effectiveness of the proposal.

**Questions:**

Please answer the questions in the above weakness section.

**Ethical Concerns:**

["NO or VERY MINOR ethics concerns only"]

**Final Justification:**

After carefully reviewing all the responses from the authors, I think the paper does present a new and effective attacking method against SAM2. I tend to raise my score to accept.

**Limitations:**

Yes

**Quality:**

3

**Strengths And Weaknesses:**

Strength: The paper is clearly written and quite easy the follow. The motivation is clear the source codes are provided. Experimental comparison with recent methods is provided and reported results demonstrate quite clear performance improvements.

Weakness:
1.	The authors claim that the proposed method generates “Universal Adversarial Perturbation”. This might not be appropriate. In adversarial attack scenario, UPA usually means a single perturbation that can be added to any input for attack. However, according to the method description, it seems that for different inputs, the perturbations are different.
2.	The authors claim that the proposal is “the first cross-prompt attack” against SAM2. This may not be appropriate either, considering that there are previous attacks against SAM, although not exactly SAM2, such as https://doi.org/10.3390/app14083312, https://doi.org/10.1145/3688636.3688653.
3.	The fundamental equation in Definition 2.1 seems wrong. To attack the model, the IoU between model outputs and ground-truth labels should be minimized instead of maximized.

---

> ### Author Rebuttal · Authors · 2025-07-30
>
> ## Responses to Reviewer HAPs ####
>
> Thank you very much for your support and suggestions on our work! You can find the responses to your questions as follows:
>
> ------
>
> **Q1**: Concern regarding the UAP
>
> **A1**: We fully agree with the reviewer’s definition of Universal Adversarial Perturbation (UAP). Our method indeed generates a single, input-agnostic perturbation that is applied uniformly across different videos and frames. All three proposed attacks—Semantic Confusion, Feature Shift, and Memory Misalignment—optimize a shared UAP. We explicitly define the adversarial example as the target frame plus the UAP in the method section (L165–166).
>
> ------
>
> **Q2**: Concern regarding the cross-prompt attack
>
> **A2**:  We agree with the reviewer that several existing works have explored cross-prompt (or prompt-free) attacks against SAM, such as [1, 2], as well as more recent efforts like DarkSAM (NeurIPS 2024) [3] and UAD (CVPR 2024) [4], which target segmentation foundation models under prompt-free settings. We already discuss this line of work in the manuscript (L34–36). However, as shown in Fig. 1, these SAM-specific attacks fail to effectively transfer to SAM2, likely due to the introduction of its memory-based architecture. Our work focuses on designing the first *effective* cross-prompt universal adversarial attack specifically for SAM2, rather than adapting existing SAM attacks. We will revise the manuscript to clarify this distinction more precisely.
>
> ------
>
> **Q3**: Concern regarding the Definition 2.1
>
> **A3**:  Thank you for pointing this out! The use of "max" is indeed a typo. We will correct it to "min" in the revised version.
>
>
>
> Reference:
>
> [1] Segment Shards: Cross-Prompt Adversarial Attacks against the Segment Anything Model, Applied Sciences 2024.
>
> [2] Cross-Prompt Adversarial Attack on Segment Anything Model, ICCBN 2024.
>
> [3] Darksam: Fooling segment anything model to segment nothing, NeurIPS 2024.
>
> [4] Unsegment Anything by Simulating Deformation, CVPR 2024.

---

> > ### Comment · Reviewer_HAPs · 2025-08-06
> >
> > Thank the authors for answering my questions in the rebuttal. After carefully reading the authors' responses to all the reviews, I think the paper does present a new and effective attacking method against SAM2. Also, the paper can be improved after the additions presented in the rebuttals. I would like to raise my rating to 'accept'.

---

> > > ### Author Response · Authors · 2025-08-06
> > > **Official Comment by Authors**
> > >
> > > Thank you very much for your thoughtful feedback and for taking the time to carefully consider our responses. We truly appreciate your recognition of the novelty and value of our work and your willingness to raise the score. We will incorporate the suggested improvements into the final version to further strengthen the paper.

---

### Official Review · Reviewer_VCeY · 2025-06-27

**Clarity:** 3
**Significance:** 3
**Originality:** 2
**Rating:** 5
**Confidence:** 3

**Summary:**

This paper addresses the problem of creating adversarial examples for the Segment Anything Model 2 (SAM2), a prompt-conditioned video segmentation model. The authors claim to introduce the first cross-prompt universal adversarial perturbation (UAP) that effectively fools the model with an architecture-specific optimization scheme. Following the standard white-box (full access) iterative procedure, this method optimizes a three-term loss function, with each term addressing a different aspect of the adversary: semantic attack aims to flip the prediction entirely from foreground to background in each frame, feature shift attack maximizes the distance of the perturbation to a prototype derived from the original image in embedding space, while the memory misalignment attack attempts to do the same across consecutive frames. The method's design is based on a preliminary analysis of SAM2's strengths and weaknesses, together with a comparison to SAM-specific baselines. UAP-SAM2 is extensively evaluated on image and video segmentation tasks using three datasets, with additional exploratory and ablation studies. Moreover, the authors investigate the influence of two defense strategies, model pruning and data preprocessing, on the method's performance.

**Questions:**

1. My current score is a conservative choice — I would have no problem increasing it once the authors provide a broad and detailed analysis of the first weakness that shows their method's novelty. Please focus on that during the rebuttal.

2. Line 42: do the authors mean percentage points?

3. Equation 2: since we want to decrease mIOU, shouldn't it be a minimum instead of a maximum?

4. Equation 7: am I right that the formula is lacking an additional sum over positive pairs?

5. How is $J_{ma}$ computed on image segmentation tasks, given that no consecutive frames are present? Is it simply skipped?

6. Appendix, line 67: what do the authors mean by _negative sample number_? Does that relate to the prototype construction and the number of used augmentations? I believe this name was not defined anywhere.

**Ethical Concerns:**

["NO or VERY MINOR ethics concerns only"]

**Final Justification:**

My main concern in the initial review was the lack of an adequate placement of the main contribution of the work (the proposed loss function) within the context of the related literature. The authors properly discussed this issue in the rebuttal, which made the contribution clearer and explained its novelty. As my initial score was a conservative choice because of this issue (which I mentioned in the review), I decided to raise it following the authors response. Regarding other smaller issues, I am satisfied with the response and hope that all of the suggestions will be included in the final version of the paper if it gets accepted.

**Limitations:**

"yes"

**Paper Formatting Concerns:**

Typos:
- lines 203-204
- line 85 (repeated "as")

**Quality:**

3

**Strengths And Weaknesses:**

Strengths:

- the paper is well-written, with a very thought-out structure. One could argue that attributing the ineffectiveness of prior (SAM-suited) works to SAM2 to archiectural and task differences is trivial. However, the authors do a very good job of carefully analyzing the influence of its different components before introducing their own method. Since it would be rather difficult to build an extensive theory for such a model, I really appreciate the intuition-based preliminaries. The notation and description of different loss components are meticulous, making it easy for the reader to understand the motivation and goals.

- the experimental evaluation seems very strong. The authors included many different scenarios, additional ablations, and investigated two scenarios of defending against their attack. In its current form, the paper provides clear evidence that UAP-SAM2 is a very effective technique with advantages over prior works.

Weaknesses:

- my main concern lies in the actual novelty of the introduced loss function. While the authors properly cite prior methods related to SAM, I think the paper is missing a broad description of approaches that create UAPs or standard adversarial attacks for classical image and video segmentation tasks (or even for SAM, no prior method is formally described in this work). Overall, how is each component placed in the related works context? What is each term's 'nearest neighbour' wrt. prior research? I'm not deep enough in the field to verify this myself and must argue that every reader should be provided with sufficient knowledge to ensure that something inside the loss function is actually new, or whether it's just a combination of techniques from other papers.

- some fancy wording and PR/marketing could be removed. This is obviously a matter of personal preference, but sometimes moving from a high-level description to a more detailed one would be beneficial:
1. lines 141--142: what do the authors mean by _simultaneously enhancing background saliency to confuse the current-frame features_?
2. lines 144--146: while the overall description of the avalanche effect here is interesting, it seems unclear how the cosine similarity is computed. Is it between the embeddings from the vision encoder? Or is it done on raw frames? In any case, I believe that the authors should also include the cosine similarity between consecutive frames in the clean case, i.e., no perturbation, for fair comparison.
3. line 103: using UAP-SAM2* as a name for the sample-wise version of UAP-SAM2 is not yet clear to me. Is the _universality_ of UAP-SAM2 meant in the context of different videos, where a single perturbation fools the model on all of them? If yes, then the sample-wise version loses its _universality_, so the naming becomes ambiguous. Obviously, this is only a small detail.

- another detail is that, in my opinion, the caption of each figure should be self-contained and self-explanatory. For example, Figure 5.'s caption is very limited, the plot's font size is very small, and by looking only at the figure it is unclear what it shows. Same goes for Figure 3. Once again, these are details.

---

> ### Author Rebuttal · Authors · 2025-07-30
>
> ## Responses to Reviewer VCeY
>
> Thanks very much for your insightful suggestions on our work! You can find the responses to your
>
> questions as follows:
>
> ------
>
> **Q1**: Concern regarding the proposed loss function
>
> **A1**:  We appreciate the reviewer’s concern about the novelty of the proposed loss function. We clarify that the significant architectural differences between SAM-series models and classical segmentation models, such as SAM’s reliance on prompt-based guidance instead of category labels, render existing adversarial attacks for standard segmentation models or classifiers largely ineffective. Recent SAM-specific attacks [1,2,3], which we cite and discuss as the most relevant baselines for our problem setting and method design, widely demonstrate this. We outline their core algorithmic ideas in the Introduction and Related Work sections (Lines 31–36, 298–306). We will gladly expand these descriptions in the revised manuscript, incorporating more explicit summaries of existing UAP-generation techniques.
>
> As we highlight in the Introduction (L37-43), our key motivation is to understand why existing SAM attack methods fail on SAM2 and to design novel losses based on its unique architecture, particularly the memory mechanism, based on the empirical findings in Section 3.1. Therefore, our method is not a simple combination of prior techniques. In the Methodology section, we explicitly credit the relevant inspirations for each loss component (L161–163, L178–180).
>
> Specifically: 1) The semantic confusion loss $J_{\text{sa}}$ uses a BCE formulation to model foreground–background ambiguity in SAM2’s prompt-based masks; 2)The feature shift loss $J_{\text{fa}}$ is motivated by meta-learning and targets semantic misalignment between the target and reference frames; 3)The memory misalignment loss $J_{\text{ma}}$ leverages our discovery of a “semantic avalanche” effect to disrupt temporal consistency across frames.
>
> ------
>
> **Q2**: Concern regarding the background saliency
>
> **A2**: Thanks for pointing this out. We mean that the added perturbation simultaneously (1) suppresses the semantics of the foreground by pushing positive mask regions toward negative values, and (2) enhances the saliency of the background by optimizing negative mask regions toward a target negative value. This contrastive manipulation strengthens the background features and weakens the foreground features, thereby improving the overall attack effectiveness on the current-frame representations.
>
> ------
>
> **Q3**: Concern regarding Fig.3
>
> **A3**: Thank you for the question. We compute the cosine similarity based on the embeddings extracted from the vision encoder. To ensure a fair comparison, we include the results for clean samples (i.e, no perturbation) under the same setting as Fig. 3 in Table R1. These results show that although the semantic content of each frame gradually shifts over time, both neighboring frames and the current frame maintain relatively high similarity to the first frame. This further supports the presence of the avalanche effect triggered by adversarial frames.
>
> Table R1: A study on the semantic feature similarity between consecutive frames
>
> |       Frame      | 0 |   1   |   2   |   3    |  4    |   5    |   6    |  7    |   8    |   9    |
>
> |----------------------------|---|----------|----------|-----------|---------|-----------|----------|---------|----------|----------|
>
> |  Benign_Adjacent | 1 | 0.9674 | 0.9710 | 0.9645 | 0.9640 | 0.9666 | 0.9746 | 0.9739 | 0.9767 | 0.9790 |
>
> |   Benign_First    | 1 | 0.9674 | 0.9545 | 0.9423 | 0.9310 | 0.9249 | 0.9191 | 0.9109 | 0.9086 | 0.9042 |
>
> | Adv_ Adjacent | 1 | 0.8736 | 0.8228 | 0.8253 | 0.8338 | 0.8337 | 0.7892 | 0.7886 | 0.7858 | 0.7718 |
> |   Adv_ First   | 1 | 0.8884 | 0.8828 | 0.7564 | 0.6492 | 0.5964 | 0.5671 | 0.5379 | 0.5171 | 0.5080 |
>
> ------
>
> **Q4**: Concern regarding UAP-SAM2*
>
> **A4**: Your understanding is correct—“universality” in UAP-SAM2 refers to the ability of a single perturbation to fool the model across different video scenes. The sample-wise setting, in contrast, generates one perturbation per sample and thus loses this universality. In adversarial attack literature, it is common to study both universal and sample-wise perturbations, so we extended UAP-SAM2 to the sample-wise setting and referred to it as UAP-SAM2* for differentiation. We acknowledge the potential confusion this naming may cause. We will adopt a clearer name in the revised version to avoid ambiguity.
>
> ------
>
> **Q5**: Concern regarding the figure caption
>
> **A5**: Thank you for the suggestion. We will revise the captions of Figures  to make them more self-contained and informative, and also improve the readability of the plots in the updated version.
>
> ------
>
> **Q6**: Concern regarding the Line 42
>
> **A6**: Yes, we report the results in percentage points to account for the difference in robustness between the target models SAM and SAM2.
>
> ------
>
> **Q7**: Concern regarding  the Eq. 2
>
> **A7**: Thank you for pointing this out! The use of "max" is indeed a typo. We will correct it to "min" in the revised version.
>
> ------
>
> **Q8**: Concern regarding  the Eq. 7
>
> **A8**: Thank you for the insightful comment. You are correct—the summation over positive pairs is missing in Equation (7). We will correct this in the revised version.
>
> ------
>
> **Q9**:  Concern regarding the Jma in image segmentation tasks
>
> **A9**: The reviewer’s understanding is correct. As demonstrated in our code, due to the absence of consecutive frames in the image dataset, we skip Jma during the optimization of the attack.
>
> ------
>
> **Q10**: Concern regarding the negative sample number
>
> **A10**:  hank you for the clarification request. The term “negative sample number” refers to the negative samples used in Equation 7 to minimize the feature distance from the target sample. We will provide a more detailed explanation in the revised version.
>
> ------
>
> **Q11**: Concern regarding typos
>
> **A11**:  Thank you for pointing this out. We will make the corresponding changes in the revised version.
>
> ------

---

> > ### Comment · Reviewer_VCeY · 2025-08-01
> >
> > Thank you for a detailed response, the answers resolved my concerns. Following the initial review, I would like to raise my score as the proposed loss function's novelty is now clearer (Q1.). I am certain that including parts of the response in the final version of the paper will improve the clarity and novelty of its main contribution.

---

> > > ### Author Response · Authors · 2025-08-04
> > > **Official Comment by Authors**
> > >
> > > Thank you for your thoughtful review and for taking the time to re-evaluate our work. We greatly appreciate your updated score and are glad that our response helped clarify the novelty of the proposed loss function. We will incorporate the relevant parts of our response into the final version to enhance the clarity and presentation of our contributions.

---

### Official Review · Reviewer_q4bX · 2025-07-03

**Clarity:** 2
**Significance:** 2
**Originality:** 3
**Rating:** 5
**Confidence:** 4

**Summary:**

This work introduces universal adversarial perturbations against SAM2 which is the first adversarial attack for SAM2. The work finds that existing adversarial attacks for SAM cannot effectively transfer to SAM2 models due to their inherently different design. The proposed attack has three components – semantic confusion, feature shift  and memory misalignment. Experiments show the effectiveness and good transferability of the proposed attack.

**Questions:**

Q1: Why setting the UAP perturbation level to 10/255 while keeping the sample-wise variant attack to 8/255? Can you show the attack performance under a range of different perturbation levels?

Q2: Figure 6(f) describes the attack performance with different settings of prompt numbers, however it seems model SAM2-T and SAM2-S exhibit completely different trend. Can you provide some explanations to this difference?

Q3: The proposed attack has three different components with unique purposes. Will these three loss components introduce difficulties in the optimization procedure? Can you show the curve of each loss components during adversarial sample optimization? From the formulation in Eq. (3), seems the scales of each loss component are equal. In practice, how do you set the loss scale to balance each module?

**Ethical Concerns:**

["NO or VERY MINOR ethics concerns only"]

**Final Justification:**

Thanks to the authors for their efforts in addressing my concerns, as well as my peer reviewers. After reading the rebuttal, I am satisfied with the further discussion and information provided by the authors. Therefore I decided to raise my rating to accept.

**Limitations:**

Yes.

**Paper Formatting Concerns:**

In Line 59, ‘m’ regions should be wrapped in math environment ($m$ regions). The font used in Figure 2 is not unified. The information provided in Figure 5 can be merged into Table 2 for a clearer comparison with different attack methods.

**Quality:**

3

**Strengths And Weaknesses:**

Strength: The proposed method is original and effective. Analysis of the previous attack failure modes provides insight. Experiments are thorough. The transferability performance is good.

---
Weakness:

W1: The perturbation bound is fixed to a certain number (Line 201) and lacks performance comparisons between different settings of perturbation levels.

---

> ### Author Rebuttal · Authors · 2025-07-30
>
> ## Responses to Reviewer q4bX ##
>
> Thank you very much for your valuable suggestions on our work! You can find the responses to your questions as follows:
>
> ------
>
> **Q1**:  Concern regarding  perturbation budgets
>
> **A1**: We follow standard practices in the adversarial attack literature for setting perturbation budgets: 8/255 for sample-wise attacks [1,2,3] and 10/255 for UAPs [4,5,6]. As these values are widely adopted in prior work, we use them as the default settings for the two attack variants in our main experiments.
>
> Moreover, we already provide an extended analysis of the proposed attack under varying perturbation levels in the supplementary material (Sec. D, L63–66), and report the results in Fig. A1(b).
>
> ------
>
> **Q2**: Concern regarding  Fig. 6(f)
>
> **A2**:  We fully understand the reviewer’s concern. The results in Fig. 6(f) are accurate and reflect real experimental observations. We believe the differing trends between SAM2-T and SAM2-S arise from architectural differences: SAM2-S has a deeper network and more parameters, which likely gives it stronger capacity to interpret multi-point prompts. With more prompt points, SAM2-S can more accurately localize the target regions, thus reducing the effectiveness of the attack.
>
> ------
>
> **Q3**: Concern regarding the curve of each loss component
>
> **A3**: Thank you for the insightful suggestion. We provide the optimization curves (in numerical form) for the three loss components during adversarial sample generation in Table R1, which reports their values at each epoch. We follow the same experimental setup as in Section 4.4 of the manuscript and conduct the experiment on SAM2-T for consistency. As shown, all losses consistently decrease and converge as training progresses, indicating stable optimization.
>
> As stated in Eq. (3) and implemented in our released code, we do not apply any additional tricks or manual weighting to balance the loss terms—their scales are treated equally. We acknowledge that better loss balancing could potentially improve performance, and we will explore this in future work.
>
>
>
> Table R1: Investigation of loss dynamics for the proposed modules during adversarial optimization
>
> |  epoch  |   0   |   1   |   2   |   3   |  4   |   5   |  6   |   7   |  8   |   9   |
>
> |-------------|---------|---------|---------|---------|--------|---------|---------|---------|---------|---------|
>
> | L_sa | 0.7468 | 0.6938 | 0.6938 | 0.6942 | 0.6946 | 0.6943 | 0.6943 | 0.6952 | 0.6954 | 0.6954 |
>
> |  L_fa | 0.2264 | 0.1625 | 0.2002 | 0.1889 | 0.1581 | 0.1968 | 0.2541 | 0.1892 | 0.2113 | 0.1665 |
>
> |  L_ma | -0.8013| -0.8101| -0.8166| -0.8272| -0.8267| -0.8287| -0.8378| -0.8440| -0.8448| -0.8439|
>
> ------
>
> **Q4**: Concern regarding formatting and presentation issues
>
> **A4**: We appreciate the reviewer’s insightful suggestions. We will address the issues raised by revising Line 59 to wrap ‘m’ regions in a math environment, unifying the font in Figure 2, and merging Figure 5’s information into Table 2 for clearer comparison in the revised version.
>
>
>
> Reference:
>
> [1] Unsegment Anything by Simulating Deformation, CVPR 2024.
>
> [2] Segpgd: An effective and efficient adversarial attack for evaluating and boosting segmentation robustness, ECCV 2022.
>
> [3] Cross-Prompt Adversarial Attack on Segment Anything Model, ICCBN 2024.
>
> [4] Darksam: Fooling segment anything model to segment nothing, NeurIPS 2024.
>
> [5] Pre-trained adversarial perturbations, NeurIPS 2022.
>
> [6] Universal adversarial perturbations, CVPR 2017.

---

### Decision · Program_Chairs · 2025-09-17

**Decision:**

Accept (spotlight)

**Comment:**

This paper introduces the first universal adversarial perturbation attack against SAM2. The proposed methodology is well-motivated and demonstrates strong performance. The authors have thoroughly addressed all reviewer concerns in their rebuttal, leading to a consensus for acceptance. As the highest-rated paper in my batch of 13 items, I recommend it for spotlight.